# Spatio–Temporal Variations in Impervious Surface Patterns during Urban Expansion in a Coastal City: Xiamen, China

**Wang Man, Qin Nie \*, Lizhong Hua, Xuewen Wu and Hui Li**

Department of Spatial Information Science and Engineering, Xiamen University of Technology, Xiamen 361024, China; manwang@xmut.edu.cn (W.M.); lzhua@xmut.edu.cn (L.H.); 2010112001@xmut.edu.cn (X.W.); lihui@xmut.edu.cn (H.L.)
\* Correspondence: nieqinhongyi@163.com

**Abstract:** Impervious surfaces (IS) coverage is a quantifiable environmental indicator for understanding urban sprawl and its potential impacts on sustainability of urban ecological environments. Numerous studies have previously demonstrated global and regional IS variation, but little attention has been paid to the different internal and external patterns of IS development as urbanization progresses. This study estimates IS coverage in a subtropical coastal area of Xiamen, southeastern China, from Landsat TM/OLI images obtained in 1994, 2000, 2004, 2010, and 2015, and quantifies its spatio–temporal variations using IS change trajectories and radar graphs. During the study period, IS gradually expanded along the shoreline in a pattern resembling the shape of the bay. The land surfaces are classified into four zones: IS1 and IS2, dominated by cultivated land and forest; IS3, complex land use/coverage; and IS4, built-up areas. The progression and transformations of these zones highlight the main trends in IS changes in the study area. The trajectories of the zones form a layered structure in which the urban centers of each district progressively gain IS4, and transformations into IS3 and IS2 extend successively beyond the centers. The orientation of IS expansion in each of the six districts of Xiamen is revealed by radar graphs. The areas containing intermediate and high percentages IS each expanded in generally consistent directions throughout the study period, except in Tong'an district, which showed a change in the direction of expansion of its area of intermediate and high IS.

**Keywords:** impervious surfaces; urban expansion; spatio–temporal variation; change trajectory; radar graphs

---

## 1. Introduction

Worldwide urbanization has progressed with unprecedented speed over the past 50 years as the percentage of the population residing in urban areas has increased from 30% in 1950 to 54% in 2014 [1]. China has experienced rapid urbanization over the past two decades [2]. Data from the World Bank East Asia and the Pacific Urban Flagship Study show that in China, the total built-up area rose from 92,151 km$^2$ in 2000 to 118,763 km$^2$ in 2010, an increase of 29%. A growing body of literature has documented rapid urban expansion in China [3–6].

One consequence of rapid urban expansion is the replacement of natural vegetation coverage with impervious surfaces (IS) such as buildings, paved roads, and parking lots. IS patterns therefore indicate the degree of urbanization, and can reveal significant information about urban areas; they can be utilized to quantify urban development and monitor urban growth [7–9]. On the other hand, IS are anthropogenic features through which water cannot infiltrate the soil, which can alter the natural

hydrological condition by increasing the volume and rate of surface runoff and decreasing ground water recharge and base flow, and affect urban surface temperatures by altering the sensible and latent heat fluxes that exist within and between the urban surface and boundary layers. IS patterns thus are also major indicators of environmental quality because IS growth presents serious challenges in terms of the environment, climate, population health, and natural resources [10–12]. Timely and accurately estimation of IS spatial distributions and exploration of spatio–temporal dynamics will thus be key to understanding the process of urban expansion and dealing with these challenges.

Since the 20th century, increasing attention has been paid to LUCC (Land Use/Coverage Change) associated with urban expansion. In contrast to land use/cover type data, IS is a major component of urbanized regions, and can better characterize intra-class information for a given land-cover type. Furthermore, land use types are defined somewhat subjectively. The land use/cover data system employed in most previous studies can only detect inter-class information regarding different land cover types, which may not represent actual land use development. In comparison, the study of a physical property of land cover (i.e., IS) has greater significance. An understanding of the changes in spatial distributions between and within land cover types can reveal comprehensive information regarding urban expansion [9,13]. Given these potential advantages for the study of urban environments, remote sensing of IS has been performed in recent years and quantification of the spatial and temporal changes in IS growth is attracting increasing interest [14].

In the past few years, numerous studies on global and regional IS variation have been made [15–27]. Table 1 lists the methodology used, along with major research conclusions and main representative references in the last 5 years. The existing researches mainly focused on (1) the depiction of IS spatial distribution and quantitative temporal comparison of IS spatial dynamics; (2) IS time series analysis over time; (3) developing models to extract or predict IS distributions from time series remote sensing data; (4) IS change pattern based on spatial metrics. To date, a general trend towards increasing IS associated with urbanization has been captured in different studied cases. All these provide planners and urban manager with a wealth of information in the planning processes. However, IS dynamic is the result of a complex system intertwined with interactions of environmental, social, and economic factors, and it is difficult to assess and compare these methods. Especially, there is no well-documented approach for analyzing the different internal and external patterns of IS development as urbanization progresses. There is a general lack of research on the development of methods and indicators to quantify and study spatio–temporal IS variation.

**Table 1.** Summary of the research on impervious surfaces (IS) in the last 5 years.

| Methodology | Images Used | Contents | References |
|---|---|---|---|
| Visual interpretation; image comparison | Multi-temporal and multi-source remote sensing images | Mapping IS dynamics | Shahtahmassebi et al. [9]; Zhang et al. [24] |
| Time series | Multi-temporal remote sensing data | IS spatio–temporal changes over time | Zhang et al. [19]; Xu et al. [20] |
| Modeling | Time series remote sensing data | Developing model to extract or predict IS distributions | Zhang et al. [22]; Li et al. [21] |
| Spatial metrics | Multi-temporal images | IS change pattern | Nie et al. [13]; Man et al. [25] |

The purpose and innovation of this study is to introduce the methods of IS change trajectory and radar graphs to analysis the different internal and external patterns of IS variation due to urban sprawl over long time periods, taking the coastal city of Xiamen, China, as the study area. The primary objectives of this work include: (1) understanding the characteristics of spatio–temporal IS variation associated with LUCC; (2) quantifying the progression of the variation (for the period 1994–2015)

using IS change trajectories; and (3) exploring the orientation of IS expansion in the context of rapid urbanization.

## 2. Methodology

### 2.1. Study Area and Data Preprocessing

The subtropical coastal city of Xiamen in Fujian province, southeastern China, was selected as the study area (Figure 1). The city's six districts cover the urban nucleus of Xiamen Island (Siming and Huli districts) and the nearby mainland (Haicang, Jimei, Tong'an, and Xiang'an districts). As one of the four earliest established special economic zones in China, Xiamen has experienced rapid urban growth and dramatic land use changes in recent decades, especially since the implementation of trade reform and liberalization policies. Consequently, the natural land cover has been changed into anthropogenic IS, especially within the core urban area of Xiamen Island. From 2000, Xiamen city underwent a transformational period of rapid urbanization, and some decision-making factors have encouraged IS coverage to sprawl its space off the island. The spatio–temporal variation of the IS pattern in Xiamen has representative significance for other urbanizing cities.

Five cloud-free Landsat TM/OLI images covering a 21-year period (1994, 2000, 2004, 2010, and 2015) were the primary data sources for mapping IS. The TM/OLI images were obtained from China Geospatial Data Cloud [28]. Atmospheric influence was removed using the MODTRAN4-based FLAASH module of the ENVI4.7 software. Image-to-image registration among the five images was conducted with a second-order polynomial transformation and nearest neighbor re-sampling. The overall root mean square (RMS) error values were less than 15 m (0.5 pixels).

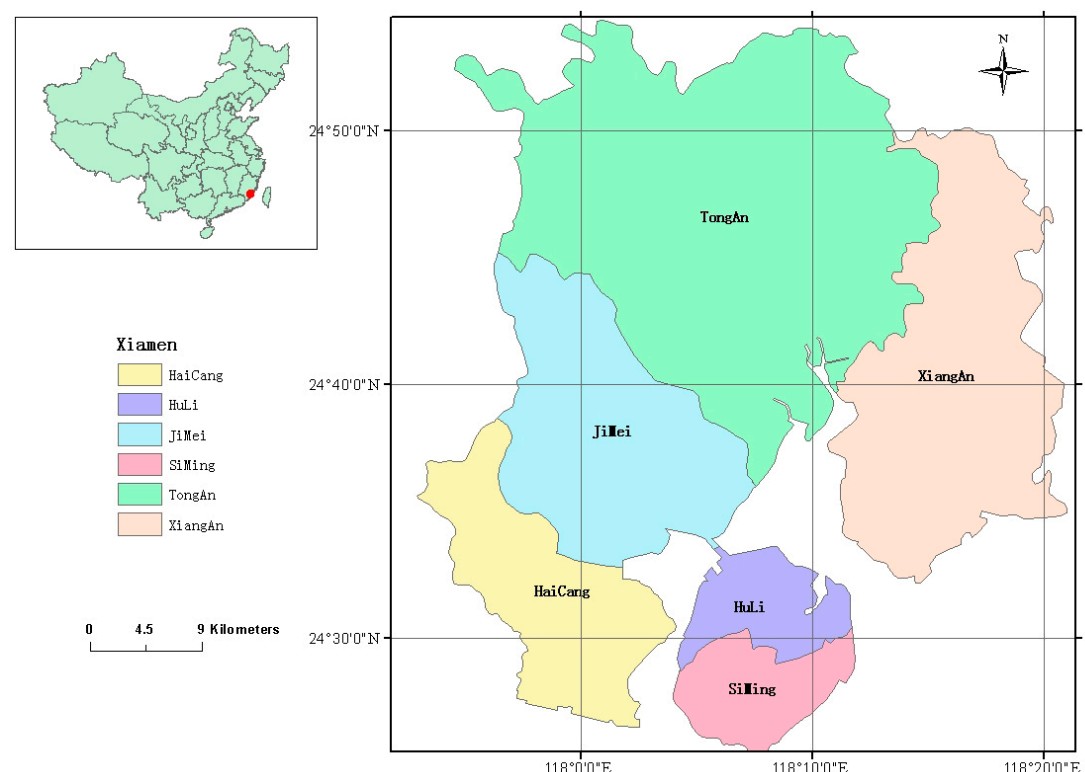

**Figure 1.** Study area.

### 2.2. Estimation of IS

Numerous remote sensing approaches have been developed to extract IS information. This study used linear spectral mixture analysis (LSMA) to estimate the spatio–temporal patterns of IS in Xiamen for each study year.

The analysis assumes that the measured spectrum is a linear combination of the spectra of all the components (endmembers) within a pixel, and can be expressed as follows [29]:

$$R_i = \sum_{k=1}^{n} f_k R_{ik} + ER_i \tag{1}$$

where $i$ is the number of spectral bands and $k$ is the number of endmembers (here $k = 4$). $R_i$ is the spectral reflectance of band $i$ containing all endmembers, $f_k$ is the proportion of endmember $k$ within the pixel, $R_{ik}$ is the known spectral reflectance of endmember $k$ within the pixel on band $i$, and $ER_i$ is the error for band $i$.

A constrained least-squares solution is used, assuming that the following two conditions are simultaneously satisfied [29]:

$$\sum_{k=1}^{n} f_k = 1 \; and \; 0 \le f_k \le 1 \tag{2}$$

The RMS error images are used to determine the overall error of all the endmember abundance values for each pixel. A low RMS error of the abundance images is desired. The RMS is calculated as follows:

$$RMS = \sqrt{\sum_{i=1}^{m} ER_i^2 / m} \tag{3}$$

Four types of endmember were selected in the present research: vegetation (grass and trees), high-albedo (metal, new concrete surfaces, sand, and some compound materials), low-albedo (old concrete surfaces, cyan tiles, and asphalt), and soil. The sum of the high- and low-albedo fractions was calculated to obtain IS fraction images.

### 2.3. IS Change Trajectory

The IS change trajectory method was introduced to quantify the spatial and temporal variability of IS changes during the study period. To achieve this objective, the imperviousness estimate was further classified into four ranks based on the IS fraction (ISF): (1) IS1 (ISF < 30%); (2) IS2 (30% ≤ ISF < 50%); (3) IS3 (50% ≤ ISF < 70%); and (4) IS4 (ISF ≥ 70%).

In general, the change trajectory of a time series can be expressed by trajectory codes in various ways for each pixel in a raster image. To capture the spatio–temporal variability of IS patterns and their change trajectories within the study area, we define here the IS change trajectory as follows:

$$\text{Trajectory} = \text{classify } 1994 \times 10{,}000 + \text{classify } 2000 \times 1000 + \text{classify } 2004 \times 100 + \text{classify } 2010 \times 10 + \text{classify } 2015 \tag{4}$$

where, Trajectory, with no mathematical sense, is the trajectory code of a given pixel; and classify 1994, classify 2000, classify 2004, classify 2010, and classify 2015 represent the IS class (1–4) in the respective year at the given pixel. Each digit within the five-digit trajectory code is thus the IS type for that single pixel at a certain time. The code therefore represents the change in IS type over the study period. For example, a trajectory value with five equal digits (e.g., 44,444) indicates no change in IS type over the study period, whereas different values within a code (e.g., 12,234) indicate that the IS type changed during the study period. The IS trajectories at each pixel were acquired to trace the history of IS changes across the study area. Raster calculation allowed a distribution map of all the trajectories in the study area to be created.

### 2.4. Radar Graphs

Radar graphs can effectively characterize the spatial direction of urban expansion. In the present research, we selected the six districts of Xiamen for analysis using radar graphs. The government office of each district in 1994 was chosen as the center of the district, and 16 fans were then drawn

from each center by extending equally spaced rays (22.5° apart). Radar graphs were then obtained by summarizing the IS area in each fan.

## 3. Results

### 3.1. Variation of Impervious Surface Association with Land Use During the Study Period

The IS fraction, which was estimated within a continuous range of between 0 and 1, was mapped for the study area. Figure 2 shows the spatial and temporal IS distributions obtained from the TM/OLI images as IS changes, for each of the five years in the 21-year study period. There was a general trend of IS expansion during the study period. In 1994 and 2000, IS distribution was sparse in the mainland districts and highest in the built-up areas on the west of Xiamen Island. Since 2000, IS increased rapidly on both the island and the mainland. On the island, IS expanded in a northward direction, covering a large area and becoming concentrated in clusters by 2015. In the four mainland districts (Haicang, Jimei, Tong'an, and Xiang'an), IS increased outward from their built-up areas, and high IS growth occurred from 2004 to 2015 on the part of the coast facing Xiamen Island: in 2004, there were scattered areas of high IS around the bays surrounding Xiamen Island and some small patches of high IS in the mountainous areas; in 2010, the high IS had extended outward from the 2004 pattern, and new high-IS spots appeared along the bay; by 2015, a more continuous pattern of high IS had emerged. The image statistics showed that the area of IS4 increased 2.4 times between 1994 and 2015, from 144.07 to 347.8 km$^2$. The area of IS3 also increased over this period, by 225.43 km$^2$, equivalent to a growth rate of 10.73 km$^2$/year. The increases in IS3 and IS4 were at the expense of IS1 and IS2, which sustained continuous decreases from 1994 to 2015. In particular, IS1 decreased from 341.35 km$^2$ in 1994 to 91.86 km$^2$ in 2015, equivalent to a rate of −11.88 km$^2$/year.

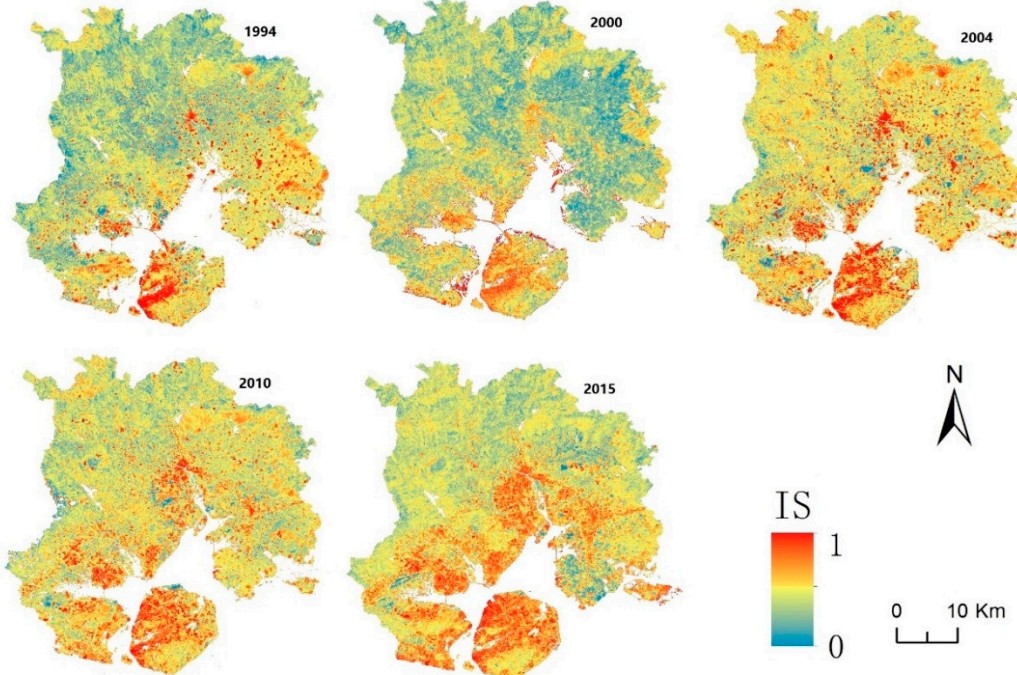

**Figure 2.** IS maps of Xiamen, China, during the period 1994–2015.

Overall, the spatial pattern of IS expanded gradually along the bay during the study period, eventually matching the coastline. Temporally, the greatest increased in IS coverage occurred from 2000 to 2015.

To better understand the IS dynamics, we mapped LUCC over the study period using the maximum likelihood procedure. We identified four types of LUCC (forest, cultivated, construction, and unutilized) and investigated these in each IS zone.

In all five study years, the main land use types in the IS1 and IS2 zones were cultivated, forest, and unutilized land (Figure 3). The percentage of construction land in the IS3 zone was greater than that in the IS1 and IS2 zones; construction land predominated in the IS4 zone, accounting for 46.19% in 1994, 79.37% in 2000, 38.42% in 2004, 86.31% in 2010, and 91.84% in 2015. The LUCC composition of the IS4 zone was more complex in 2004 than in the other study years. Unutilized land covered 32.4% of the total IS4 area in that year. Comparison of the high-resolution image with data from the Statistical Yearbook suggests that a dramatic change in LUCC occurred in about 2004, and many scattered construction and unutilized land could be found in the mountain area.

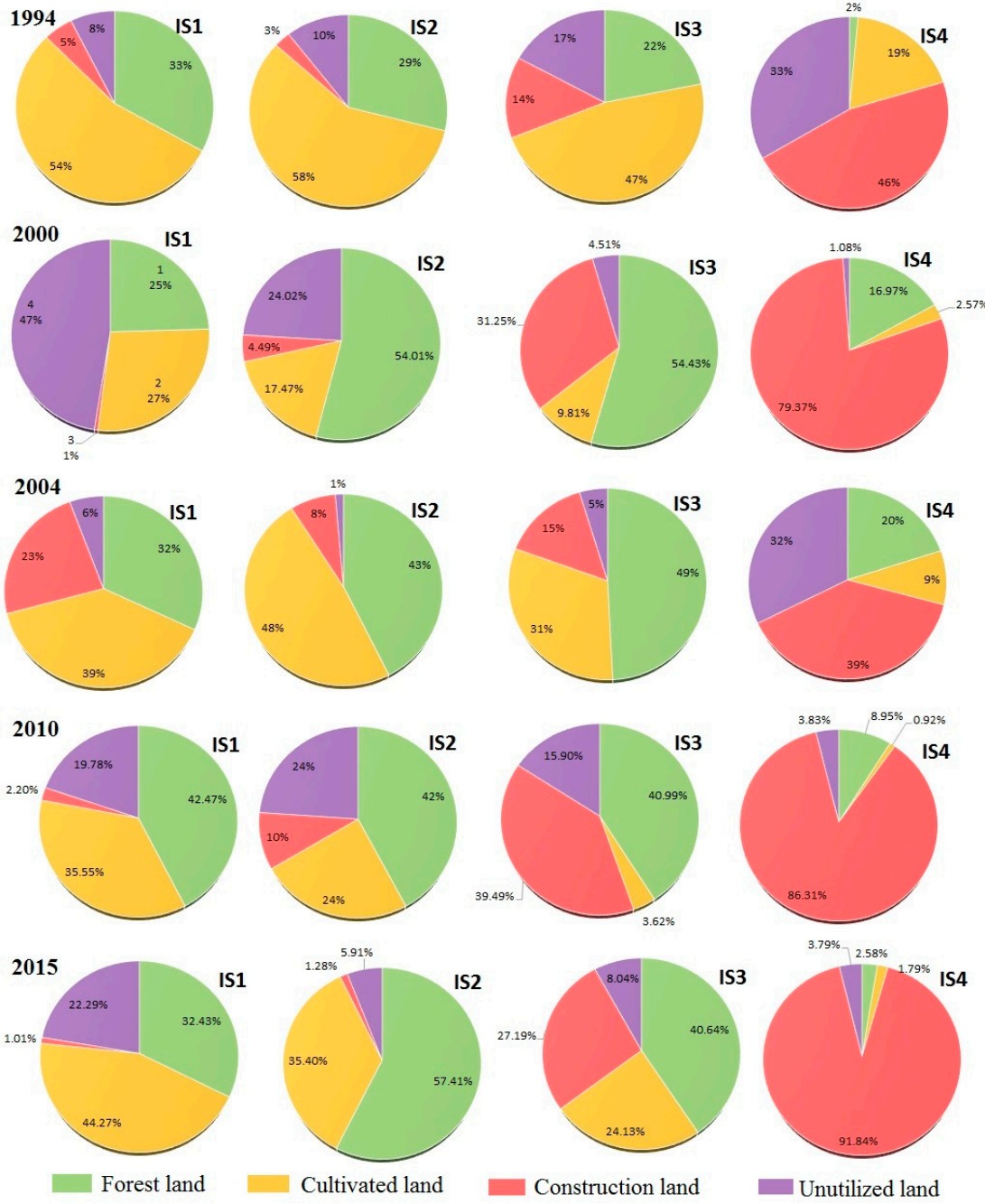

**Figure 3.** Land use/coverage change (LUCC) compositions in each IS zone.

### 3.2. Progression of IS Spatio–Temporal Variation

The visual interpretation and image statistics in the above analysis reveal expansion of IS and an increasingly contiguous spatial pattern. In this section we quantify the spatio–temporal progression of IS variation, using IS change trajectories.

The IS change trajectories describe the progression of spatio–temporal change in IS during the study period. There were 1018 possible IS change trajectories during the study period. After excluding erroneous trajectories, we counted the number of trajectories that transformed into different IS types, and then calculated the area for each (Table 2). There were 255 trajectories (****4) that transformed into IS4, covering 20.96% of the study area, of which 7.01% were 44,444 trajectories. The number of trajectories that transformed into IS3 and IS2 was 256 for both, accounting for 35.32% and 37.48%, respectively, of the whole study area. IS change trajectories that transformed into IS1 covered the smallest percentage of the study area (6.24%). Therefore, the trajectories that transformed into IS4, IS3, and IS2 depicted the main change trends in IS type for the study area.

Figure 4 shows the spatial patterns of these IS transformations. The different types of trajectory form a layered structure on the map. Those transforming into IS4 lied mainly in the built-up areas of all six districts, and formed the center of the structure. The next layer was mainly ****3 trajectories located around the ****4 trajectories. On their fringes lied the ****2 trajectories. Extending further, the ****1 trajectories were distributed in the outermost layer. The layered structures of IS change trajectories highlighted the spatial and temporal evolution of urbanization in this coastal city; i.e., urban sprawl started around the bay area, and then extended outward.

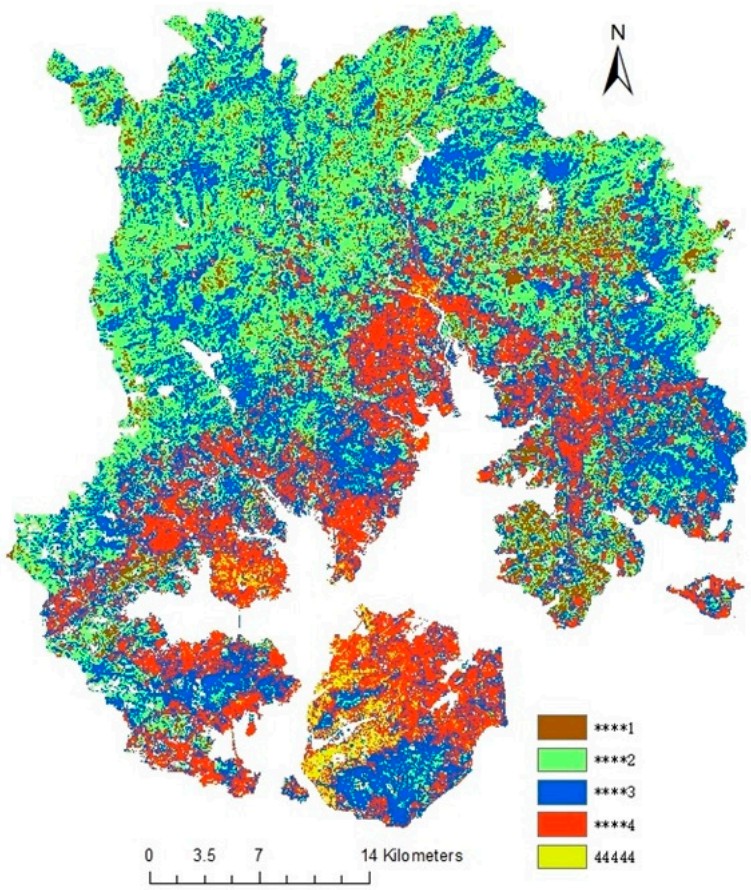

**Figure 4.** IS change trajectories for the period 1994–2015.

**Table 2.** Statistics of IS change trajectories in the study area.

| Trajectory Codes | Pixels Number | Percentage (%) |
|:---:|:---:|:---:|
| ****1 | 94,523 | 6.24 |
| ****2 | 568,169 | 37.48 |
| ****3 | 535,472 | 35.32 |
| ****4 | 295,445 | 19.49 |
| 44,444 | 22,281 | 1.47 |

Notes: * refers to the IS type code that is the integer value from 1 to 4; ****1 indicates the trajectory transforming into IS1 from 1994 to 2000, 2004, 2010, and 2015; ****2 indicates the trajectory transforming into IS2, and so on.

### 3.3. Spatio–Temporal Orientation of IS Expansion Associated with Urban Expansion

As shown above, the areas of low IS (IS1; IS values of 0–30%) were located mainly in areas covered by vegetation. The areas of intermediate IS (IS2 and IS3; IS values of 30–70%) were located at the fringes of urban areas. The areas with the highest IS (IS4; with IS values > 70%) were located in the inner urban areas. Areas having intermediate and high percentages of IS were most closely connected with human activity, and are thus discussed in detail, with particular emphasis on the highest IS areas. The radar graphs in Figures 5 and 6 show the orientation characteristics of urban expansion according to the direction of IS spread for each area.

In Siming district, the area with highest IS percentages extended mainly north–northeast, northeast, and east–northeast by 2015, having extended north–northeast, northwest, and east–northeast in 1994 and 2000. Extension continued east–northeast and east in 2000–2004. Of note, the area that had the highest percentage decrease in IS during 2004–2010 continued to sprawl east–northeast and east in 2010 and 2015. The area with intermediate percentages extended east–northeast in 1994–2015, but the total area decreased in 2010 and 2015.

In Huli district, the highest IS area extended mainly west and west–southwest in 1994 and 2000, and extended west, west–southwest, and northwest in 2004 and 2010. This trend of expansion continued until 2015. The area with intermediate percentages decreased during 1994–2015. Extension was mainly west–southwest, west–northwest, east–northeast, and east in 1994, before extending in 2000 to the northwest and north–northwest, but with a decreasing area in the east and east–northeast. By 2004, there were decreases in area in all directions, which continued gradually until 2010, with a sharp decline in 2015.

In Jimei district, the high IS extended mainly east and northwest in 1994. It then expanded gradually westward from 1994 to 2015. The intermediate IS grew slightly northwest and west–northwest during the whole study period.

In Tong'an district, the extensions of high IS varied in a complicated manner. Following a period of little change in 1994–2000, there was a sudden large change in 2004, toward west–northwest, northwest, north–northwest, and north–northeast. However, these areas then decreased in all directions except the south in 2010, and turned to extend south, southwest, south–southwest, and south southeast in 2015. The area with intermediate IS kept expanding to west–northwest, west, northwest, north–northwest, and north over the study period.

In Xiang'an district, the high IS extended mainly north, north–northeast, northeast, and east–northeast over the study period, but decreased in area in all these directions in 2000. However, the area expanded greatly to the north, north–northwest, northwest, north–northeast, and east–southeast by 2015. The area with intermediate percentages gradually expanded north, north–northeast, northeast, and east–northeast during the study period.

In Haicang district, the area with high percentages IS extended mainly west–northwest, then northwest, west–southwest, and southwest during 1994–2004. It decreased slightly by 2010, before showing sharp extensions in all these directions by 2015. The area with intermediate percentages grew slightly west–northwest, then west and northwest during the study period.

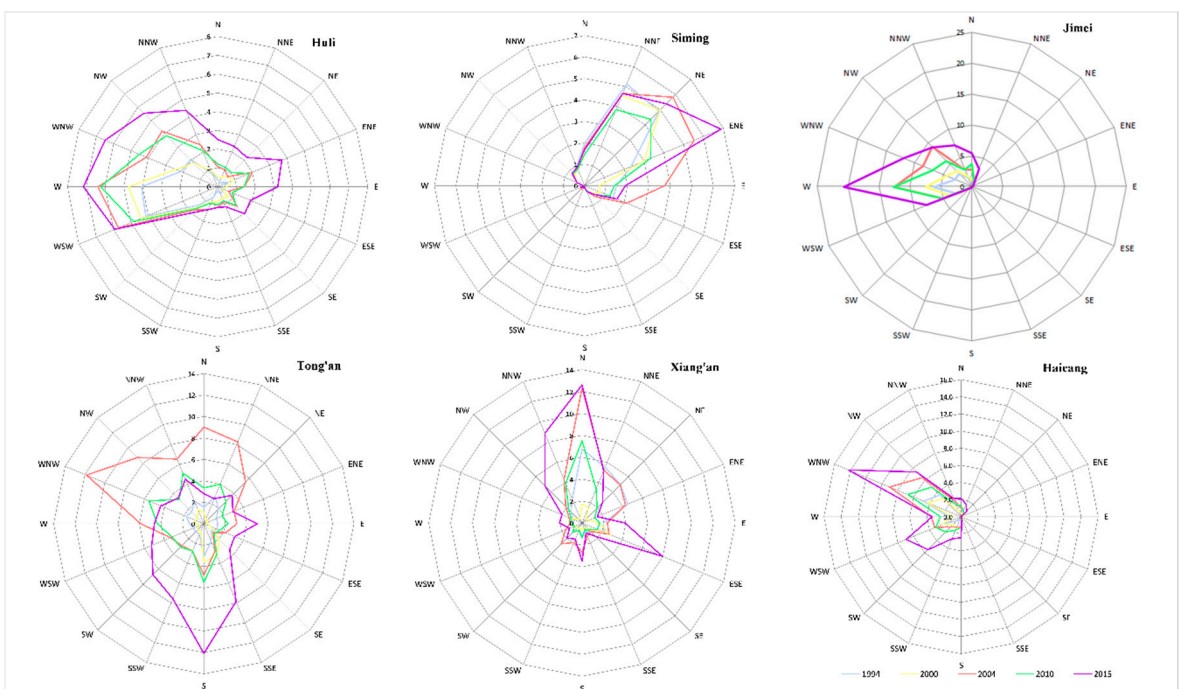

**Figure 5.** Spatial orientation of the expansion of high percentages IS in the period 1994–2015.

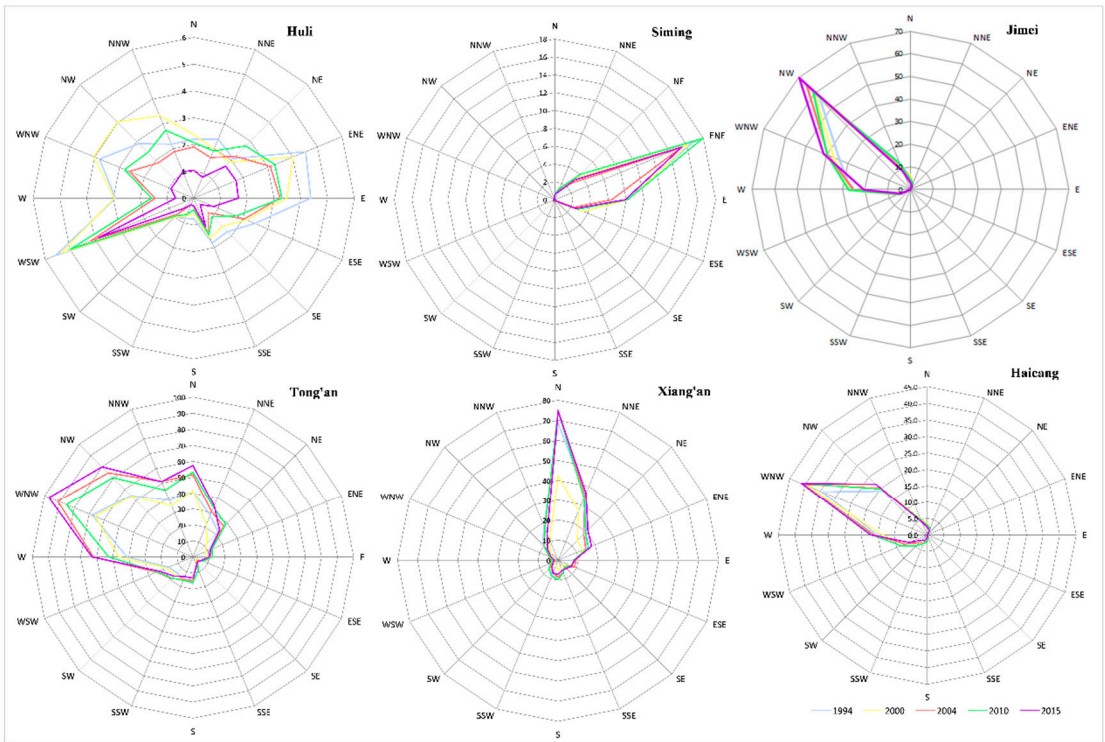

**Figure 6.** Spatial orientation of the expansion of intermediate percentages IS in the period 1994–2015.

## 4. Discussion and Conclusions

### 4.1. Discussion

This article introduces IS change trajectory to explore the spatial and temporal changes of IS, thereby giving a deeper understanding of the temporal dynamics of urbanization in a rapidly developing coastal city. The IS change trajectory can be used to examine the way in which IS evolves

spatially and temporally. Radar graphs are also employed to reveal the orientation characteristics of urbanization and urban sprawl.

Spatio–temporal change in IS is driven by natural, economic, and political factors. Among the various factors leading to urbanization of the present study area, economics and politics have played important roles. As a modern coastal city, Xiamen's urbanization now extends beyond Xiamen Island and along the nearby mainland coast. The reforms and policies following China's opening to trade in 1978 led initially to a rapid increase in economic activity in Xiamen city, and accompanying large-scale changes in land use and coverage. Since the 1990s, Xiamen Island, which is the center of Xiamen city, has seen increased construction in the Siming and Huli districts. From 1990 to 2000, Xiamen city underwent a transformational period of rapid urbanization from being solely an island city to having development extend along the coast facing the island. In 2005, the urban master plan for Xiamen city was revised and a policy was adopted to optimize land use on the island and expand urban space off the island. Urban expansion has shifted to beyond Xiamen Island, with intensive construction along the mainland coast. High fraction IS has thus spread rapidly along the coast. Therefore, the IS pattern in the study area mirrors economic development and government policy.

The analysis of IS spatio–temporal variation can provide insights to aid the successful planning and management of cities. Numerous documented studies have indicated that the spatio–temporal expansion pattern of IS is an important basic ingredient for analyzing urbanization evolution from pattern to process. IS variation can inform policy on smart growth strategies, and improve our ability to assess and create future planning scenarios for decision-makers. As reported by Sterling and Ducharne in 2008, anthropogenic land cover has totaled up to ~40% of the Earth's surface and caused environmental degradation in the world. Of particular concern in many urbanizing regions is the increasing IS, which has resulted in less precipitation, more dryness and higher temperatures. A large amount of studies on the fields of biogeochemical cycles and urban climate have mapped IS coverage for the requirement of urban sustainable planning and environmental protection. For example, Franco Salerno et al. [30] suggested that reduction in IS was a suitable strategy to adapt to severe worsening of river water quality in northern Italy by limiting the construction of new impervious areas and decreasing the existing areas by only 15%. Xu et al. [31] found that the variation in the proportions of IS in the Xiong'an New Area, China could result in a significant shift of LST, and a balance in the amount of total IS area in the 2000 km$^2$ new area is 433 km$^2$. Exceeding/reducing the amount would result in a rise/decline of the area's LST. Yan et al. [32] found that percentage of total IS area can thus be used as a key criterion for urban planning to ameliorate urban biodiversity, of which above 40%, plant diversity decreased sharply, and the proportion of exotic species rose. Similar kinds of studies on IS can provide many other insights [33,34].

### 4.2. Conclusions

Information on IS spatio–temporal variation in the context of rapid urbanization is key to understanding the process of urban expansion and deal with these serious challenges in terms of the environment, climate, population health, and natural resources. This study applied IS change trajectories and radar graphs from multi-temporal Landsat imagery to investigate the spatio–temporal change in IS patterns. A general trend of increasing IS in Xiamen, China was found during the study period. Of note, there has been rapid expansion of IS both on and beyond Xiamen Island since 2000. Spatially, the IS growth along the bay appeared gradually from 1994 to 2015, eventually leading to an IS pattern that matches the coastline of the bay. Construction land predominated in the IS4 zone, whose percentage was also greater in the IS3 zone than that in the IS1 and IS2 zones.

The trajectories of IS transforming into IS types IS4, IS3, and IS2 highlight the main trends in IS changes in the study area. The spatial patterns of the different types of trajectory from 1994 to 2015 form layered structures that highlight the way in which urbanization evolves spatially and temporally. Radar graphs reveal the spatial orientations of IS expansion. The high fraction IS in all six districts showed consistent extension in certain directions throughout the study period, except in Tongan district,

which showed a change in the direction of expansion from north to south after 2010. These change progressions and orientations in IS patterns during the study period show the urbanization evolution from pattern to process in Xiamen, China, which thus provides useful information for urban sustainable planning and environmental protection.

**Author Contributions:** Q.N. and W.M. conceived, designed and performed the experiments; Q.N. and L.H. analysed the data; X.W. and H.L. contributed materials/analysis tools; W.M. and Q.N. wrote the paper.

**Funding:** This research was funded by the National Natural Science Foundation of China (41501447, 41471366, and 31670645); the Fujian Natural Science Foundation, China, (2017J01666); and the Program for New Century Excellent Talents in Fujian Province University.

**Acknowledgments:** The authors would like to express special thanks to the anonymous reviewers for their constructive suggestions and comments.

**Conflicts of Interest:** The authors declare no conflict of interest.

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
