# Peer review of "Spatio–Temporal Variations in Impervious Surface Patterns during Urban Expansion in a Coastal City: Xiamen, China"

_sustainability, doi:10.3390/su11082404_

Round 1

Reviewer 1 Report

The paper is interesting and can add to the body of literature on ways to measure sprawl.  Overall, I think that discussion on the usefulness and practical application of this research can be expanded greatly. Below are more specific comments.

Page numbers and line numbers are missing making it hard to dictate where my comments are referring to.

The abstract should refer to sprawl, specifically. Also, it should tell the reader why we care about IS. What's the significance to sustainability related issues?

Intro

Can a definition be provided for built up area?

We need more information on the significance of IS - water quality, storm water management, air quality, urban heat island, etc.

You state that IS is more effective than "traditional methods of studying urban growth, which are based on land use/cover change (LUCC)". Please give a sentence or 2 contrasting the 2 methods. This is touched on in the discussion, but should be introduced to the reader in the introduction.

Study area and data reprocessing should move to methodology.

Where did the tm/oli images come from?

Methods

Caption for figure 2 needs more details - what does a 1 and 0 mean? Figure legends should be close to self explanatory without reference to the paper.

Figure 3 caption - define acronyms

More details in all captions necessary.

Header should read Discussion and Conclusion

Last sentence of the discussion is really important. Can the authors expand on this and talk about why using IS techniques can aid in this? How can this data be applied for practical purposes? To inform policy on smart growth strategies? To identify areas in need of more trees/vegetation to manage water and air quality, etc? Is there any political will to contain sprawl and manage growth in a thoughtful way? It might help to compare and contrast to ways that other cities around the world managed rapid growth - some did it in more efficient, sustainable, health conscious ways than others did. Perhaps comparing some Nordic countries to US cities in the Southwest, for example. 

Author Response

Point 1: Page numbers and line numbers are missing making it hard to dictate where my comments are referring to.

Response 1: According to the suggestion, we added the page numbers and line numbers. Please see the revised manuscript.

Point 2: The abstract should refer to sprawl, specifically. Also, it should tell the reader why we care about IS. What's the significance to sustainability related issues?

Response 2: According to the suggestion, we revised the abstract and added urban sprawl and the significance of IS study. Please see page 1.

Point 3: Intro

Can a definition be provided for built up area?

Response 3: Build-up area here means urban. We replaced it with urban in manuscript.

Point 4: We need more information on the significance of IS - water quality, storm water management, air quality, urban heat island, etc.

You state that IS is more effective than "traditional methods of studying urban growth, which are based on land use/cover change (LUCC)". Please give a sentence or 2 contrasting the 2 methods. This is touched on in the discussion, but should be introduced to the reader in the introduction.

Response 4:  According to the suggestions, we added more information on the significance of IS, and placed the contrasting of LUCC with IS to introduction portion. Please see P2 L13-20, L23-28 and P3 L1-7.

Point 5: Study area and data reprocessing should move to methodology.

Where did the tm/oli images come from?

Response 5: According to the suggestion, we moved section 2 into methodology section, and added the description about sources of TM/OLI images in the manuscript. Please see  section 2.1 in page 4.

Point 6: Caption for figure 2 needs more details - what does a 1 and 0 mean? Figure legends should be close to self explanatory without reference to the paper.

Figure 3 caption - define acronyms

More details in all captions necessary.

Response 6:  For figure 2, o and 1 refer to the range of IS fraction, and we added the explanation about this in our manuscript. Please see page 7 line 16-17.

For figure 3, we added the information about figure 3 acronyms (LUCC) in page 2 Line 23-24.

Point 7: Header should read Discussion and Conclusion

Last sentence of the discussion is really important. Can the authors expand on this and talk about why using IS techniques can aid in this? How can this data be applied for practical purposes? To inform policy on smart growth strategies? To identify areas in need of more trees/vegetation to manage water and air quality, etc? Is there any political will to contain sprawl and manage growth in a thoughtful way? It might help to compare and contrast to ways that other cities around the world managed rapid growth - some did it in more efficient, sustainable, health conscious ways than others did. Perhaps comparing some Nordic countries to US cities in the Southwest, for example. 

Response 7: According to the suggestion, we revised the header as Discussion and Conclusion, and expand the discussion about why using IS can aid successful planning and management of cities. Please see page 16.

Reviewer 2 Report

In the last few years many research have been published about impervious surfaces in China.

It is not clear to understand the novelty of this artcicle.

There are two main weaknesses:

a) bibliography is too much old and not enough developed, please develop with more recent references

b) it is not clear, compared with other research, what is new, please make a summary table of the research on IS in China in the last 5 year comparing methodologies and strengths and weaknesses of different research and the novelty fo this arrticle 

Author Response

Point 1: bibliography is too much old and not enough developed, please develop with more recent references.

Response 1: According to the suggestion, we reviewed literatures again and added some latest references. Please see references section.

Point 2: it is not clear, compared with other research, what is new, please make a summary table of the research on IS in China in the last 5 year comparing methodologies and strengths and weaknesses of different research and the novelty fo this article.

Response 2: According to the suggestion, we added a research summary on IS variation and presented the novelty for our study. Please see paragraph 2-4, page 3.

Reviewer 3 Report

The paper attempts to bring the Radar Graph component as a novelty to explain the IS change trajectories. Neither the usage of Radar graph to explain urban LULC expansion is new nor the IS change trajectories are new. However, using different IS classes and the change trajectories in conjunction with Radar graph to explain the spatio-temporal variation can be helpful in urban sustainability studies. The paper needs to address how the present study could contribute to the literature in this regard. It needs to be significantly modified with references to urban sustainability studies that focuses on these topics. Furthermore, it needs to address how this research is going to contribute in the urban sustainability field. The discussion and conclusion portion needs to reflect these points.  Currently, it feels that as if the paper is written for a remote sensing focused journal. But there are plenty of literature available in remote sensing journals that is very similar to this paper. 

Author Response

Point 1: using different IS classes and the change trajectories in conjunction with Radar graph to explain the spatio-temporal variation can be helpful in urban sustainability studies. The paper needs to address how the present study could contribute to the literature in this regard. It needs to be significantly modified with references to urban sustainability studies that focuses on these topics. Furthermore, it needs to address how this research is going to contribute in the urban sustainability field. The discussion and conclusion portion needs to reflect these points.

Response 1: Thanks for the constructive suggestions. According to the suggestions, we retrieved literatures again and added them to reference, and we revised the discussion and conclusion portion. Please see page 16.

Reviewer 4 Report

The paper aims at measuring spatio-temporal variations in impervious surface patterns during urban expansion in Xiamen, China. This study explores and important issue. The structuring of the manuscript is ok but the presentation lacks proper sequencing or flow.

 The main weakness of the paper is to appear as a technical exercise that may be sound in itself but is not related to the field of study it pretends to address. This is obvious in the way the introduction is written (“state of the art”), in the absence of discussion about urban growth.

The authors seem to be unaware of the state of knowledge in the domain they investigate (neither fractal analysis nor existing simulation models as those for instance by Guy Engelen and Roger White or Michael Batty are quoted) and they pretend to bring useful instruments for monitoring urban growth in a rather naïve way. In my opinion it deserves a second chance if a major revision is made that includes the comments given below.

Introduction

-          It is not clear what research questions the authors sought to answer; please, clarify the specific objectives of the paper; highlight the aspects in which your research departs from the existing literature, and what are its innovative contributions to science.

-          I would encourage the authors to directly compare this method with other common methods as this would be very useful to the broad community. More discussion of issues and alternative approaches could be helpful.

-          To my view introducing the paper as a very general review of urban expansion, it is taking the topic from a too long distance. In other words, the literature which the paper is confronting is not the right one. There are plenty of papers measuring the spatial dimension of urban growth that should be quoted instead of these too general references.

-          I would encourage the author to introduce a new text regarding previous work on urban expansion measurement. You could deeply review the work of Jaeger and Ewing...

And

1)      Application of a new GIS tool for urban sprawl in Europe (2015), Forum Wissen, 57-64.

2)      Urban permeation of landscapes and sprawl per capita: New measures of urban sprawl (2010), Ecological Indicators 10 (2), 427-441.

3)       Assessing the Effect of Spatial Proximity on Urban Growth. Sustainability 2018, 10, 1308.

4)      “Measuring Sprawl and Its Impacts: An Update,” Journal of Planning Education and Research, 35(1), 2015, 35-50

Methodology

-          Study area: The authors should be clear about why choosing this particular region in China.

Results

-          There are very tedious descriptions of analytical results in the paper but no real discussion of theoretical elements.

Conclusions/ discussion: 

-          Can the authors suggest how the employment of this analysis can help decision makers coordinate planning policies? Or any suggestions how this analysis can be used to develop instruments for implementation of policies?

-           

-          According to the conclusion section, I think the paper is just a case study. It doesn’t have the clear research problem.

-           

-          Please, highlight the main (innovative) achievements/conclusions of the research.

Author Response

Point 1: Introduction: It is not clear what research questions the authors sought to answer; please, clarify the specific objectives of the paper; highlight the aspects in which your research departs from the existing literature, and what are its innovative contributions to science.

I would encourage the authors to directly compare this method with other common methods as this would be very useful to the broad community. More discussion of issues and alternative approaches could be helpful.

 Response 1: According to the suggestion, the introduction portion has been carefully revised. We refined the research object and innovations. Please see paragraph 2~4 in page 3.

 Point 2:  To my view introducing the paper as a very general review of urban expansion, it is taking the topic from a too long distance. In other words, the literature which the paper is confronting is not the right one. There are plenty of papers measuring the spatial dimension of urban growth that should be quoted instead of these too general references.

  I would encourage the author to introduce a new text regarding previous work on urban expansion measurement. You could deeply review the work of Jaeger and Ewing...

And

1)      Application of a new GIS tool for urban sprawl in Europe (2015), Forum Wissen, 57-64.

2)      Urban permeation of landscapes and sprawl per capita: New measures of urban sprawl (2010), Ecological Indicators 10 (2), 427-441.

3)       Assessing the Effect of Spatial Proximity on Urban Growth. Sustainability 2018, 10, 1308.

4)      “Measuring Sprawl and Its Impacts: An Update,” Journal of Planning Education and Research, 35(1), 2015, 35-50.

Response 2: According to the suggestions, we added such literatures that measure the spatial dimension of urban growth. Please see reference portion.

Point 3: Study area: The authors should be clear about why choosing this particular region in China.

Response 3: According to the suggestion, we added the reason why choosing this particular region in China. Please see page 4 line 7-15.

Point 4: There are very tedious descriptions of analytical results in the paper but no real discussion of theoretical elements. Can the authors suggest how the employment of this analysis can help decision makers coordinate planning policies? Or any suggestions how this analysis can be used to develop instruments for implementation of policies?

According to the conclusion section, I think the paper is just a case study. It doesn’t have the clear research problem. Please, highlight the main (innovative) achievements/conclusions of the research.

Response 4: According to the suggestions, we added the discussion of how IS spatio-temporal variation can provide insights to aid the successful planning and management of cities. Please see the second paragraph in page 16.

Round 2

Reviewer 2 Report

I'm not conpletely satisfied.

I asked for a table summarizing existing literature and novelty of the arcticles (can be a result!)

Many articles are published on this topic, and a new article shoul be able to adjunt something important.

Author Response

Point 1: I asked for a table summarizing existing literature and novelty of the arcticles (can be a result!) Many articles are published on this topic, and a new article shoul be able to adjunt something important.

Response 1: According to the suggestion, we added a research summary on IS variation. Please see table 1, page 4.

Reviewer 3 Report

Please modify the conclusion so that it reflects the objective of the paper from a urban sustainability perspective.

Author Response

Point 1: Please modify the conclusion so that it reflects the objective of the paper from a urban sustainability perspective.

Response 1: According to the suggestions, we revised the conclusion portion. Please see page 17.

Reviewer 4 Report

Thank you and your colleagues for the modifications that you have made to this article and how well you have responded to the suggestions.

Author Response

Thank you!